# SerenDiff: Generating Serendipity Recommendations through a Diffusion Model

## Abstract

Serendipity means an unexpected but valuable discovery. It has attracted wide attention in recommender systems research in recent years. Due to its elusive and subjective nature, serendipity is difficult to model even with today's advances in machine learning and deep learning techniques. In addition, most existing serendipity models lack interpretability. To address the modeling challenges and the interpretability issues, we propose a **seren**dipity **diff**usion recommendation model (named *SerenDiff*), to generate serendipity recommendations leveraging a state-of-the-art generative AI model, the diffusion model. We regarded a user history with a recommender system as an "image", and the serendipity recommendation generation as a recovering process of the corrupted "image". Diffusion models are believed to be creative in the recovering process of a noised image in the sense that they base on but go beyond the original training data, providing room for finding serendipity. Extensive experiments have shown the effectiveness of *SerenDiff*. We believe *SerenDiff* will empower everyday users, not only with increased chances of encountering unexpected but relevant discoveries, but also with interpretations on the elusive serendipity recommendations.

## 1 Introduction

Serendipity is a concept associated with unexpected discoveries that are valuable. The rapid development of information technology now offers individuals more flexibility to search for and select information that matches their needs or interests, mostly immediate needs or interests. However, most traditional deep learning-based recommender models disproportionately emphasize relevance-oriented demands that dominate the observable preference distribution and overlook the serendipity area (as shown in Figure 1(a)). This problem is known as the "filter bubble" phenomenon, which initially refers to the information systems selectively focusing on the information a user would like to see and is now borrowed in recommender systems (Fu et al., 2023a). Currently, many people have expressed their hope that recommender systems could play a role in facilitating incidental exposure to information or serendipity, whereby individuals "stumble upon" unexpected but valuable information that they did not actively seek. Different from relevance-oriented models, serendipity-oriented models attempt to provide recommendations that are out of users' expectations but satisfy their demands. However, modeling serendipity in recommender systems is a great challenge due to serendipity's elusive nature. As shown in Figure 1(b), it is challenging for models to effectively identify users' serendipity demand "zones" from the vast unexplored item space. Another challenge is related to interpretability. Many existing serendipity recommendation models (Pandey et al., 2018; Li et al., 2019; 2020a; Zhang et al., 2021; Fu et al., 2023b) struggle in the inherent transparency to reveal their inner workings.

In this paper, we leverage diffusion models to address the challenges of both serendipity modeling and interpretability. Diffusion models, which have been demonstrated to be powerful generative frameworks capable of capturing complex data distributions and exploring less frequent but meaningful patterns, can therefore be leveraged to model the serendipity distribution based on user behaviors. In addition, diffusion models are believed to be creative in the sense that they can generate new and unique data beyond the original training data and demonstrate high creativity (Cao et al., 2024; Zhou & Lee, 2024). **We would like to investigate the feasibility of diffusion models to generate serendipity during the process of corrupting and recovering a user's history record with a recommender system.** Specifically, we first transform the one-dimensional user's interacted

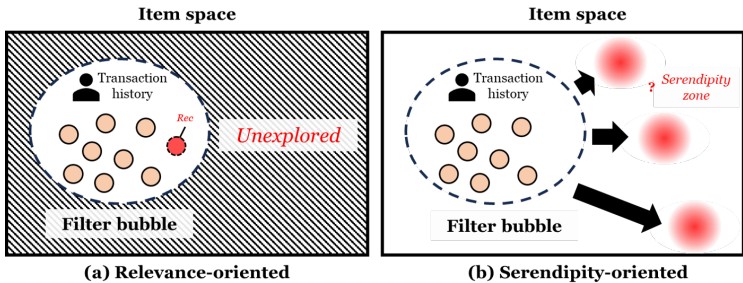

Figure 1: Relevance-oriented recommendations vs. serendipity-oriented recommendations

items into two-dimensional user portraits from the perspectives of unexpectedness and relevance, reflecting the two essential facets of the serendipity concept (Fu et al., 2023b). In the image-like user portrait, each interacted item is a point. We then adopt the conditional diffusion model (Saharia et al., 2022; Rombach et al., 2022) to generate serendipity points conditioned on the image-like user portrait, during the image recovering process from the added noises.

The proposed model, named the **seren**dipity **diff**usion recommendation model (*SerenDiff*), has two advantages. First, the diffusion and the denoising processes not only learn the user's current or immediate demands but also creatively infer the user's "zone" of hidden, dormant, or potential demands. Second, the constructed user portrait visualizes the generated serendipity "zone" from the two dimensions of unexpectedness and relevance. The distance and the direction between the "zone" and the user's history could naturally serve as a form of interpretation for serendipity. Extensive experiments on two real-world datasets and a case study have demonstrated the effectiveness of the *SerenDiff* model.

The core novelty of this paper is two-fold: 1) the adaptation of diffusion models into a new domain marked by unique challenges: serendipity recommendations, and 2) a novel user history representation as a two-dimensional image-like user portrait.

## 2 RELATED WORK

This study draws on two research lines: serendipity research as well as diffusion recommendation models.

### 2.1 SERENDIPITY RESEARCH

First coined by Harold Walpole in 1754, the word "serendipity" is used to describe the process of making discoveries by accident, but it received little attention until the mid-1900s when it was used as a descriptor of accidental or unplanned discovery in the scientific context (Merton & Barber, 2011). Although there is some disagreement as to the precise nature of serendipity, all accounts agree that the following two aspects are central: **an unexpected but relevant discovery**. We leveraged these two aspects as the two dimensions for the user portraits.

Since 2018, a few recommender systems researchers have attempted to build deep learning models for serendipity recommendations. Examples are *SerRec* (Pandey et al., 2018), *HAES* (Li et al., 2019), *DESR* (Xu et al., 2020), *NSR* (Xu et al., 2020), *PURS* (Li et al., 2020a), *SNPR* (Zhang et al., 2021), and *SerenEnhance* (Fu et al., 2023b). Although their serendipity definition and computational implementations vary, these efforts collectively demonstrate the power of deep learning models' representation and prediction ability. However, none of them leverages diffusion models, the state-of-the-art generative AI models with high exploration and creativity and therefore high potential for generating serendipity. Also, most of these studies struggle with interpreting the serendipity recommendations after the recommendations are made, due to the non-transparency nature of deep learning models.

## 2.2 DIFFUSION MODELS IN RECOMMENDER SYSTEMS

Recently, diffusion models have attracted attention in recommender systems research. The study (Wang et al., 2023) found that the diffusion model could help recommender systems to infer users' future interaction probabilities through the process of gradually corrupting and recovering the users' historical interactions. The research (Yang et al., 2024) designed a guided diffusion model for recommendation tasks named DreamRec, which guided the denoising process using users' histories to generate recommendations aligned with the historical interactions. The paper (Li et al., 2023) proposed a diffusion-based recommendation model for item representations. They also argued that the noisy target item representation in the diffusion process also injected some uncertainty into this process, which could enhance the model's robustness. Meanwhile, the study (Liu et al., 2023) leveraged the diffusion model to solve the data sparsity problem in sequential recommendation tasks. The diffusion model generated high-quality augmented user interaction data based on the user history. The studies above demonstrate the effectiveness of diffusion models as recommendation models. However, all of them are accuracy-based recommendations. **None addresses serendipity tasks, which present new challenges for representing a user and constructing the guidance for the denoising process.**

## 3 PRELIMINARIES

We will use the conditional diffusion model (Saharia et al., 2022; Rombach et al., 2022) as the base model. The objective of a diffusion model parameterized by $\theta$ is to model the data generation process from noises. We will recap the basics of the diffusion model with $T$ diffusion steps for the data generation tasks, involving the forward and reverse processes.

- **Forward Process.** Let $x_0 \sim q(x_0)$ denotes an input data sample (typically a 3D tensor for an image), the forward process perturbs the input data by progressively adding a Gaussian noise with a variance schedule $[\beta_1, \beta_2, .., \beta_T]$:

$$q(x_{1:T}|x_0) = \prod_{t=1}^{T} q(x_t|x_{t-1}),$$
$$q(x_t|x_{t-1}) = \mathcal{N}(x_t; \sqrt{1-\beta_t}x_{t-1}, \beta_t\mathbf{I}), \tag{1}$$

where $t \in 1, .., T$ denotes the diffusion step and $\mathcal{N}$ refers to the Gaussian noise distribution. The hyperparameter $\beta_t$ in $\mathcal{N}$ is to schedule the noise scale added at each step $t$. After the forward process, the input data sample $x_0$ will be completely perturbed to $x_T$, which approximates to a standard Gaussian distribution $\mathcal{N}(0, \mathbf{I})$. The added Gaussian noise of each step will serve as the ground truth of the reverse (denoising) process.

- **Reverse Process.** Starting from the completely perturbed data sample $x_T$, the reverse process aims to recover the original data sample $x_0$ by iteratively removing the added noises. The reverse process can be formulated as:

$$p_\theta(x_{0:T}) = p(x_T) \prod_{t=1}^{T} p_\theta(x_{t-1}|x_t),$$
$$p_\theta(x_{t-1}|x_t) = \mathcal{N}(x_{t-1}; \mu_\theta(x_t, t), \Sigma_\theta(x_t, t)), \tag{2}$$

where $\mu_\theta$ and $\Sigma_\theta$ are the mean and covariance of the reversed Gaussian distribution predicted by a neural network (e.g., U-Net or Transformer) with the parameter $\theta$. To maintain training stability, $\Sigma_\theta$ is commonly set as a constant, while the mean $\mu_\theta$ of the distribution is further factorized as:

$$\mu_\theta(x_t, t) = \frac{1}{\sqrt{\alpha_t}}(x_t - \frac{1-\alpha_t}{\sqrt{1-\bar{\alpha}_t}})\epsilon_\theta(x_t, t), \tag{3}$$

where $\alpha_t = 1 - \beta_t$ and $\bar{\alpha}_t = \prod_{s=1}^{t} \alpha_s$. $\epsilon_\theta$ is the neural network parameterized by $\theta$ to predict the source noise $\epsilon \sim \mathcal{N}(0, \mathbf{I})$.

- **Optimization.** The objective of the diffusion model is to minimize the differences between the noise distribution added in the forward process and the noise distribution predicted in the reverse process. The objective function of the diffusion model can be written as:

$$\mathcal{L}_\theta = \mathbb{E}_{t,\epsilon \sim \mathcal{N}(0,\mathbf{I})}[\|\epsilon - \epsilon_\theta(x_t, t)\|_2^2]. \tag{4}$$

## 4 SERENDIFF: A SERENDIPITY DIFFUSION RECOMMENDATION MODEL

In this section, we will provide the details of our proposed *SerenDiff* model. Different from most serendipity recommenders, we formulate the serendipity recommendation task as an item generation task. We make use of the diffusion models' strong generation ability to produce serendipity point(s) in a user image-like portrait.

### 4.1 PROBLEM FORMULATION

Let $I = \{i_1, i_2, \ldots, i_{|I|}\}$ represents the set of items, and $T^u = \{i_1^u, i_2^u, \ldots, i_n^u\}$ represents a user history: a sequence of interacted items for the user $u$. Our goal is to generate serendipity point(s) $\tilde{M}_{seren}^u$, conditioned on a two-dimensional user portrait $M^u$, converted from the user's history $T^u$. The task can be mathematically represented as:

$$\tilde{M}_{seren}^u \leftarrow f_\theta(M^u), \tag{5}$$

where $f_\theta$ is the *SerenDiff* model with the parameter $\theta$. $f_\theta$ generates the serendipity point(s) $\tilde{M}_{seren}^u$ by randomly sampling a pure noise matrix $\tilde{M}_{seren,T}^u \sim \mathcal{N}(0, \mathbf{I})$ as the initial state and iteratively conduct the generation process $\tilde{M}_{seren,T}^u \rightarrow \tilde{M}_{seren,T-1}^u \rightarrow \ldots \rightarrow \tilde{M}_{seren,0}^u(\tilde{M}_{seren}^u)$ conditioned on $M^u$:

$$\tilde{M}_{seren}^u \leftarrow p(\tilde{M}_{seren,T}^u) \prod_{t=1}^{T} p_\theta(\tilde{M}_{seren,t-1}^u | \tilde{M}_{seren,t}^u, M^u). \tag{6}$$

### 4.2 USER PORTRAIT CONSTRUCTION

As mentioned in Related Work, researchers agree that the following two elements are essential for the concept of serendipity: unexpectedness and relevance. Therefore, we constructed **a two-dimensional user portrait along these two aspects, explicitly representing the relationship and trade-off between the two concepts.**

Specifically, we will calculate an expectedness score and a relevance score for each item in the user's history. A low expectedness score and a high relevance score together indicate a serendipitous item. For the expectedness score, inspired by the computational method for calculating surprise levels proposed by the study (Fu et al., 2023b), we define expectedness, the opposite of surprise, as the conditional likelihood of seeing an item given a user's history. Therefore, the level of the expectedness of item $i_m^u$ in $T^u$ at time $m$ is calculated as:

$$exp = p(i_m^u | T_{[1:m-1]}^u), \tag{7}$$

where the conditional likelihood $p(i_m^u | T_{[1:m-1]}^u)$ could be seen as a user's expectation of seeing item $i_m^u$ given the history $T_{[1:m-1]}^u$. For more details of the expectedness score calculation, please refer to Appendix A. After a mathematical re-writing in the study (Fu et al., 2023b), all of the components in Equation 13 could be pre-calculated from the dataset and users' historical records.

There are many existing ways to quantify an item's relevance in the recommender research community. In this study, we choose the cosine similarity as the measure of relevance with assumption that a relevant item should be similar to at least some interacted items in the user history. This study is also open to other relevance quantification approach. Let $T_{[1:m-1]}^u = \{i_1^u, i_2^u, \ldots, i_{m-1}^u\}$ denotes a history of interacted items before time $m$. We calculate the relevance score of the item $i_m^u$ in $T^u$ at time $m$ as the maximum cosine similarity between it and any interacted item in $T_{[1:m-1]}^u$:

$$rel = \max_{\mathbf{e}_h^u \in T_{[1:m-1]}^u} \frac{\mathbf{e}_m^u \cdot \mathbf{e}_h^u}{\|\mathbf{e}_m^u\| \, \|\mathbf{e}_h^u\|}, \tag{8}$$

where $\mathbf{e}_h^u$ is the embedding vector of an interacted item $i_h^u$ in $T_{[1:m-1]}^u$. In addition, considering the cosine similarity ranges from -1 to 1, we further conducted the min-max normalization to transform it into the range of [0,1].

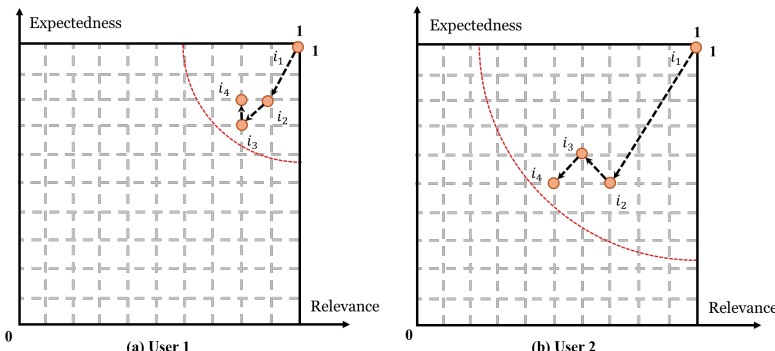

Figure 2: Two example users' interactions in their portraits

Having the approach to calculating the expectedness and relevance scores for each item in the user's history, we placed the items on the two-dimensional user portrait plane $M^u$. The first interacted item $i_1^u$ in $T^u$ reflects the user's initial preference on the expectedness and relevance level. We initialized it as (1,1) (expectedness=1, relevance=1) on the plane. For the subsequent interacted items, we placed them according to their expectedness and relevance scores. This way, we transform the one-dimensional user history sequence $T^u$ to a two-dimensional user portrait $M^u$. $M^u$ visualizes the user's preferences from the perspectives of expectedness and relevance, and also suggests the user's openness or acceptance of less expected and less relevant items. **The two-dimensional user portrait explicitly represents the trade-off between the two essential concepts through geometric distance and direction. It provides a structured and interpretable visualization that can be communicated to both model developers and users**, who appreciate recommendations that are just beyond typical choices — stretching, but not breaking a comfort zone.

As illustrated in Figure 2(a), the interacted items' radius to the anchor point (1,1) is relatively smaller, indicating that the user is conservative, less willing to try items that deviate from their niche. In contrast, the user's appetite in Figure 2(b) is bolder, more willing to explore the "outer" area. To inject the item content information into this user portrait plane, the dimensions of the item embeddings are set as the channels, just like a RGB image having 3 channels. The multi-channel user portrait image is ready to be processed by the diffusion model.

### 4.3 DIFFUSION AND DENOISING FOR SERENDIPITY GENERATION

*SerenDiff* could be described as two stages: the training stage and the inference stage. During the training stage, the model consists of two processes: the forward process and the reverse process. During the inference stage, the model only consists of the reverse process.

**Training Stage.** Figure 3 shows the training stage of *SerenDiff*. For the forward process, given a user's serendipity ground truth item(s) $M_{seren}^u$, the noises are added following Equation 1. By

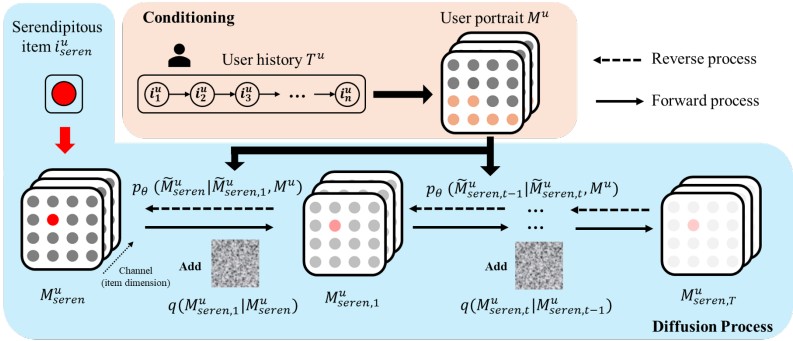

Figure 3: The training stage of the *SerenDiff* model

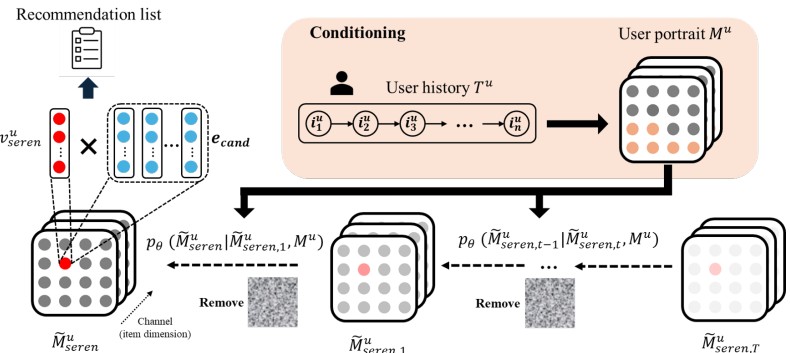

Figure 4: The inference stage of the *SerenDiff* model

adding the Gaussian noise iteratively, $M^u_{seren}$ finally degrades to the standard Gaussian noise $\mathcal{N}(0, \mathbf{I})$. In the reverse process, the *SerenDiff* model recovers the serendipity item(s) $M^u_{seren}$ from the standard Gaussian noise. Starting from the nearly pure Gaussian noise, the reverse process iteratively removes the added Gaussian noise to reconstruct the serendipity items following Equation 2.

**Inference Stage.** During the inference stage as in Figure 4, the model only has the the reverse process to recover the serendipity point(s) conditioned on a user portrait $M^u$. During the process, the next-step $\tilde{M}^u_{seren,t-1}$ is generated by removing the noise from the current noised $\tilde{M}^u_{seren,t}$ and conditioned on the user portrait $M^u$. The Gaussian noise removed by each step is predicted as:

$$\mu_\theta(\tilde{M}^u_{seren,t}, M^u, t) = \frac{1}{\sqrt{\alpha_t}}(x_t - \frac{1-\alpha_t}{\sqrt{1-\bar{\alpha}_t}})\epsilon_\theta(\tilde{M}^u_{seren,t}, M^u, t), \tag{9}$$

where $\epsilon_\theta(\tilde{M}^u_{seren,t}, M^u, t)$ is the neural network's (parameterized by $\theta$) prediction on the source noise $\epsilon \sim \mathcal{N}(0, \mathbf{I})$. Following (Saharia et al., 2022; Rombach et al., 2022), we implement $\epsilon_\theta(\tilde{M}^u_{seren,t}, M^u, t)$ using U-Net. Following Equation 4, the loss function for training $\epsilon_\theta(\tilde{M}^u_{seren,t}, M^u, t)$ can be written as:

$$\mathcal{L}_\theta = \mathbb{E}_{t,\epsilon \sim \mathcal{N}(0,\mathbf{I})}[\left\| \epsilon - \epsilon_\theta(\tilde{M}^u_{seren,t}, M^u, t) \right\|^2_2], \tag{10}$$

where $\epsilon_\theta$ is the U-Net parameterized by $\theta$ for predicting the ground truth Gaussian noise, $\epsilon \sim \mathcal{N}(0, \mathbf{I})$, added in the forward process.

### 4.4 MATCHING THE GENERATED SERENDIPITY POINT(S) TO EXISTING ITEMS

The generated serendipity point(s) in $\tilde{M}^u_{seren}$ represent the user's "zone" of serendipity, and may not match existing item embeddings in the item set $I$. To obtain the existing items, we use the cosine similarity score $s$ to search for the most similar items to the generated serendipity point(s):

$$s = \frac{\mathbf{v^u}_{seren} \cdot \mathbf{e}_{cand}}{\|\mathbf{v^u}_{seren}\| \|\mathbf{e}_{cand}\|}, \tag{11}$$

where $\mathbf{v}^u_{seren}$ (a vector) is one generated serendipity point from $\tilde{M}^u_{seren}$ (a 3D tensor). $\mathbf{e}_{cand}$ denotes the embedding of an existing item in $I$ in the same area (a grid system in the user portrait) with $\mathbf{v}^u_{seren}$. The $s$ score in Equation 11 is calculated along the embedding (channel) dimension. If the generated serendipity point's area has zero or an insufficient number of candidates, we employed the nearest neighbor search, which expands the search to adjacent areas in order to retrieve the existing item(s). Ranking by $s$, we identified a top-$k$ recommendation list, which contains the $k$ closest items to the generated serendipity point(s) for the user $u$.

## 5 EXPERIMENT SETUP

### 5.1 DATASETS

Currently, there are only two publicly available serendipity ground truth datasets with user history. One is *Serendipity 2018* (Kotkov et al., 2018), which was collected using a small-scale survey with 481 participants in the movie domain. The size of the data may not be sufficient for model training. The other one is *SerenLens* (Fu et al., 2023b), which was collected from crowd workers' annotations on the review writers' serendipity experience. It contains annotations of 265,037 book reviews and 74,967 movie reviews. The reviews and the review writers' history are from the Amazon Review Data (McAuley et al., 2015), reflecting diverse, real-world experiences for those review writers. This dataset provides a reasonable scale of ground truth data in both the book and movie domains, as shown in Table 1. Therefore, we adopt the *SerenLens* dataset, denoting it as *SerenLens-Books* and *SerenLens-Movies*.

Table 1: Key statistics of *SerenLens* (Fu et al., 2023b)

|  |  | **Books** | **Movies & TV** |
|---|---|---|---|
| **HITs (Tasks)** | Total HIT tasks assigned in MTurk | 8,268 | 4,427 |
|  | Total HIT tasks accepted in MTurk | 4,396 | 2,342 |
| **Worker Judgements** | Total worker judgements collected | 41,340 | 22,135 |
|  | Total worker judgements accepted | 21,980 | 11,710 |
|  | Judgements with initial agreement | 16,040 | 10,985 |
|  | Judgements need a third opinion | 3,960 | 725 |
|  | Degree of initial agreement | 72.98% | 93.81% |
|  | Reviews with judgements | 10,000 | 5,000 |
|  | Reviews of serendipity | 2,557 | 714 |
|  | Reviews of non-serendipity | 7,443 | 4,286 |
| *SerenCDRLens* **Dataset** | Users involved in the reviews of serendipity | 2,346 | 619 |
|  | **Total reviews involved** | **265,037** | **74,967** |
|  | Items involved in the reviews of serendipity | 2,227 | 634 |
|  | **Total items involved** | **113,876** | **23,950** |

### 5.2 EVALUATION METRICS

Since serendipity is sparse ($\approx 1.0\%$ in books and $\approx 3.0\%$ in movies) in *SerenLens*, we adopted a recall-based metric, Hit Ratio ($H_{seren}@k$). $H_{seren}@k$ measures the proportion of times a serendipity item is retrieved in the top-$k$ position (each time 1 for yes and 0 otherwise). In order to take the rank information into consideration and assign higher weights to higher ranks, we propose another metric called Serendipity-Based Normalized Discounted Cumulative Gain ($N_{seren}@k$) based on the well-known metric NDCG. $N_{seren}@k$ is calculated as:

$$N_{seren}@k = \frac{DCG_{seren}@k}{IDCG_{seren}@k} = \frac{\sum_{i=1}^{k} \frac{2^{seren_i}-1}{log_2(i+1)}}{\sum_{i=1}^{k} \frac{2^{seren_i^*}-1}{log_2(i+1)}}, \tag{12}$$

where $seren_i$ is the serendipity score (1 or 0) for the $i^{th}$ item in the rank and $seren_i^*$ is the serendipity score (1 or 0) for the $i^{th}$ item in the ideal ranking where the ranks are according to the descending $s$ (as in Equation 11) scores.

In addition to evaluating serendipity, we conducted a second set of experiments to evaluate how much sacrifice on relevance *SerenDiff* was making (if there was) in order to accommodate serendipity. Therefore, we used the two standard metrics for relevance evaluation: H@k (the "hit" here means hitting a relevant item in the top-k position) and Normalized Discounted Cumulative Gain (N@k). For all of the metrics above, a higher value indicates a better performance.

### 5.3 BASELINE MODELS

To evaluate the performance of *SerenDiff*, we selected the following three groups of representative baseline recommendation models. The first group consists of five well-known deep learning serendipity recommendation models: *DESR* (Li et al., 2020b), *PURS* (Li et al., 2020a), *SNPR* (Zhang et al., 2021), *SerenEnhance* (Fu et al., 2023b), and *SerenPrompt* (Fu & Niu, 2024). The second group is two well-known deep learning recommendation models for relevance tasks: *SASRec* (Kang & McAuley, 2018) and *BERT4Rec* (Sun et al., 2019). We applied them for the serendipity recommendation task. The third group is two recent representative diffusion-based recommendation models for relevance tasks: *DiffRec* (Wang et al., 2023) and *DreamRec* (Yang et al., 2024). We also applied them for the serendipity recommendation task. All of three groups of models are state-of-the-art recommendation models published in the top venues in recent years.

### 5.4 PARAMETER SETTINGS

We trained *SerenDiff* and the baseline models on *SerenLens*. For all of the baseline models, we reported the results using the optimal hyperparameter settings. For the proposed *SerenDiff* model, the scheduler was the linear $\beta$-schedule Ho et al. (2020).

For all the models, we adopted the 5-fold cross-validation approach to evaluate the performance. 80% of the data in *SerenLens* was used for training and the rest 20% was for testing. For each user in the test set, we held out one serendipity item as the testing positive sample, and then paired it with 999 non-serendipity items as the negative samples. To fairly evaluate the serendipity potential of all areas in the user portrait, the 1,000 candidate items in the test set were intentionally chosen to evenly distribute over the $10 \times 10$ regions in a user portrait. For more implementation details, please refer to Appendix B. On the acceptance of the paper, we will make our *SerenDiff* model publicly available through the Github website.

## 6 RESULTS

### 6.1 HYPERPARAMETER ANALYSIS

We first investigate the impact of the number of diffusion steps and the size of user portraits on model performance and efficiency. The experiments were conducted on the *SerenLens-Books* data.

**Number of Diffusion Steps.** We compared the model performance and efficiency with varying number of diffusion steps, while keeping the same channel size and user portrait size. As in Table 2, we find that increasing the number of diffusion steps steadily improves serendipity performance, while the associated increase in computational cost remains moderate and manageable. Therefore, we selected 1,000 diffusion steps for the following experiments.

Table 2: Model performance and efficiency comparison with different diffusion steps

| diffusion step | $H_{seren}$@10 | training time per batch in seconds | inference time per instance in seconds |
|---|---|---|---|
| 5 | 42.42% | 11.60s | 0.58s |
| 10 | 41.91% | 11.96s | 0.59s |
| 50 | 42.55% | 12.64s | 0.80s |
| 100 | 43.98% | 13.04s | 0.93s |
| 500 | 45.49% | 13.80s | 1.15s |
| **1,000** | 47.11% | 14.38s | 1.63s |

**Size of User Portraits.** In addition, we further compared the model performance and efficiency between the two sizes of a user portrait 10×10 and 100×100, while keeping the diffusion step 1,000. As shown in Table 3, compared with the $10 \times 10$ setting, the performance gain of the $100 \times 100$ was marginal while the efficiency dropped sharply. These results indicate that scaling up the portrait size yields diminishing returns relative to the increased computational overhead. Therefore, we set the size of user portraits $10 \times 10$. The choice of a $10 \times 10$ portrait size reflects a trade-off between computational cost and representational granularity. Larger portrait sizes substantially increase computational cost, whereas smaller ones risk oversimplifying user histories and producing

overly coarse representations, obscuring the subtle variations necessary for accurately identifying serendipity "zones".

Table 3: Model performance and efficiency comparison with different sizes of user portraits

| diffusion step | $H_{seren}@10$ | training time per batch in seconds | inference time per instance in seconds |
|---|---|---|---|
| $10 \times 10$ | 47.11% | 14.38s | 1.63s |
| $100 \times 100$ | 49.59% | 248.00s | 4.13s |

## 6.2 OVERALL PERFORMANCE

The experiment results are illustrated in Table 4. Overall speaking, our proposed *SerenDiff* achieves the best performance on both $H_{seren}@k$ and $N_{seren}@k$ at varying k levels, suggesting the power of the diffusion and denoising processes in generating serendipitous recommendations. *SerenEnhance*, which also decomposes serendipity into unexpectedness and relevance, also achieves some good performance, confirming the effectiveness of such decomposition. We observe that *SerenEnhance* outperforms *SerenDiff* in $H_{seren}@1$ and $N_{seren}@5$ metrics on the *SerenLens-Books* dataset. The reason could be *SerenEnhance*'s pairwise learning objective that has a strong ability to discriminate items. Therefore, it has a better performance in the top-1 book recommendations. Our proposed *SerenDiff*, on the other hand, shows better performance in the top-5 and top-10 recommendations due to its open-mindedness and creativity during the item-generating (or recommendation) process.

Table 4: The performance comparison of different models. The reported number is the average of 5 folds. The best results in each column are bolded and the second best results are underlined. ∗ denotes that our proposed model has statistically significant differences with at least six of the nine baseline models under a two-tailed t-test with $p < 0.05$.

| | | Serendipity-Oriented Experiments | | | | | | | | | |
|---|---|---|---|---|---|---|---|---|---|---|
| | | SerenLens-Books | | | | | SerenLens-Movies | | | | |
| | | $H_{seren}@1$ | $H_{seren}@5$ | $H_{seren}@10$ | $N_{seren}@5$ | $N_{seren}@10$ | $H_{seren}@1$ | $H_{seren}@5$ | $H_{seren}@10$ | $N_{seren}@5$ | $N_{seren}@10$ |
| Serendipity Models | DESR | 6.25% | 23.02% | 36.67% | 0.144 | 0.178 | 2.73% | 10.15% | 22.26% | 0.059 | 0.098 |
| | PURS | 5.76% | 22.60% | 32.20% | 0.139 | 0.170 | 2.44% | 6.67% | 11.54% | 0.044 | 0.057 |
| | SNPR | 7.46% | 24.09% | 38.81% | 0.149 | 0.192 | 4.27% | 16.59% | 23.90% | 0.096 | 0.112 |
| | SerenEnhance | **9.81%** | 30.49% | 45.63% | **0.329** | 0.364 | 5.07% | 19.92% | 31.25% | 0.124 | 0.161 |
| | SerenPrompt | 8.08% | 25.23% | 43.43% | 0.252 | 0.326 | 4.30% | 16.02% | 25.39% | 0.100 | 0.130 |
| Classic Rec Models | SASRec | 6.13% | 25.33% | 41.37% | 0.157 | 0.209 | 3.33% | 11.25% | 20.83% | 0.068 | 0.099 |
| | BERT4Rec | 8.03% | 27.02% | 41.79% | 0.166 | 0.214 | 5.37% | 15.28% | 20.65% | 0.106 | 0.129 |
| Diffusion-Based Models | DiffRec | 6.64% | 23.44% | 40.23% | 0.149 | 0.204 | 4.69% | 16.41% | 21.87% | 0.099 | 0.126 |
| | DreamRec | 7.42% | 24.61% | 41.02% | 0.152 | 0.208 | 5.86% | 17.19% | 27.73% | 0.116 | 0.140 |
| | *SerenDiff* | 9.03%∗ | **31.48%**∗ | **47.11%**∗ | 0.324∗ | **0.382**∗ | **6.25%**∗ | **21.48%**∗ | **33.59%**∗ | **0.134**∗ | **0.174**∗ |
| | | Relevance-Oriented Experiments | | | | | | | | | |
| | | SerenLens-Books | | | | | SerenLens-Movies | | | | |
| | | H@1 | H@5 | H@10 | N@5 | N@10 | H@1 | H@5 | H@10 | N@5 | N@10 |
| Serendipity Models | DESR | 1.92% | 6.61% | 16.63% | 0.034 | 0.072 | 0.89% | 4.69% | 9.81% | 0.027 | 0.044 |
| | PURS | 1.68% | 5.23% | 9.70% | 0.024 | 0.043 | 0.82% | 4.39% | 7.29% | 0.022 | 0.031 |
| | SNPR | 1.92% | 5.82% | 10.51% | 0.029 | 0.051 | 0.85% | 3.22% | 9.81% | 0.025 | 0.037 |
| | SerenEnhance | 2.35% | 9.81% | 20.04% | 0.042 | 0.102 | 1.04% | 6.24% | 9.85% | 0.028 | 0.052 |
| | SerenPrompt | 1.95% | 8.98% | 16.41% | 0.055 | 0.079 | 1.56% | 6.25% | 12.89% | 0.040 | 0.061 |
| Classic Rec Models | SASRec | 2.99% | 11.98% | 17.46% | 0.062 | 0.103 | 1.91% | 6.04% | 11.85% | 0.033 | 0.054 |
| | BERT4Rec | **3.13%** | **13.28%** | **23.05%** | **0.076** | **0.107** | **2.35%** | **8.20%** | 14.06% | **0.052** | **0.070** |
| Diffusion-Based Models | DiffRec | 2.28% | 8.53% | 13.62% | 0.042 | 0.067 | 1.29% | 6.84% | 10.87% | 0.029 | 0.059 |
| | DreamRec | 2.41% | 8.76% | 18.54% | 0.042 | 0.097 | 1.61% | 6.60% | **14.52%** | 0.032 | 0.066 |
| | *SerenDiff* | 2.35% | 8.74% | 15.57% | 0.039 | 0.071 | 1.62% | 6.50% | 10.65% | 0.029 | 0.058 |

In addition to serendipity-oriented evaluations, we also conducted relevance-oriented evaluations to examine the trade-offs between serendipity and relevance if any. We used the same training dataset to train the models as the first set of serendipity-oriented experiments, but for the test dataset, the ground truth was changed to relevance, available in the original Amazon Review Data (McAuley et al., 2015). As shown in Table 4, *SerenDiff* achieves better relevance than most serendipity recommendation models, but slightly worse than most classic recommendation models, demonstrating that *SerenDiff* does not substantially compromise relevance to accommodate serendipity.

## 6.3 ABLATION STUDY

We further conducted an ablation study to investigate the benefit of the two-dimensional user portrait by replacing it with two widely adopted user representation approaches in recommender systems: (1) a user embedding, represented as a single vector for each user generated from the user-item rating matrix through matrix factorization and (2) a user history sequence, represented as a sequence of item embeddings for the items in the user history. The result in Table 5 reveals that the two-dimensional user portrait has the best model performance in serendipity recommendations.

Table 5: The ablation study results of different user representations on SerenLens-Books dataset. The reported number is the average of 5 folds. The best results in each column are bolded. "↓" indicates a performance drop more than 20% relative to that of the original *SerenDiff* model.

|  | $H_{seren}@1$ | $H_{seren}@5$ | $H_{seren}@10$ | $N_{seren}@5$ | $N_{seren}@10$ |
|---|---|---|---|---|---|
| SerenDiff-User Portrait | **9.03%** | **31.48%** | **47.11%** | **0.324** | **0.382** |
| SerenDiff-User Embedding | 6.04% ↓ | 22.24% ↓ | 34.54% ↓ | 0.137 ↓ | 0.167 ↓ |
| SerenDiff-User Sequence | 7.29% | 25.63% | 39.39% | 0.153 ↓ | 0.205 ↓ |

We also investigate the time complexity and efficiency of *SerenDiff*. Results are in Appendix C.

## 6.4 CASE STUDY

To have an intuitive understanding on the results and interpretability of *SerenDiff*, we selected two example users to showcase the generated serendipity recommendations. From Figure 5(a), we observe that this user prefers the action movies. The user portrait indicates that this user tends stay well within his preference. Therefore, the serendipity point (the red point) generated by *SerenDiff* is relatively close to (1,1), successfully predicting that *Shoot to Kill*, also an action movie, is a serendipity to this user. In contrasts, the user in Figure 5(b) likes to explore various movies. Accordingly, *SerenDiff* generated the user's serendipity point (the red point) rather apart from (1,1), also successfully predicting that *Firelight*, a romance movie, is a serendipity. This movie greatly deviates from the user's usual type. The user portraits visualize the distances and directions of these serendipitous movies compared to users' typical reach, offering a form of interpretation.

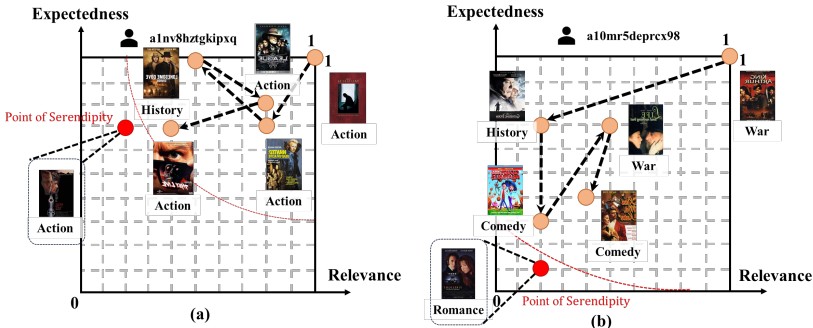

Figure 5: Two example users for a case study. The orange points denote the users' interacted items and the red points denote the generated serendipity points.

## 7 CONCLUSION

This paper contributes a new diffusion-based serendipity recommendation called *SerenDiff*, leveraging diffusion model's strong exploration and generation ability as well as high creativity. *SerenDiff* transforms a user's history into an image-like user portrait from the perspectives of unexpectedness and relevance. Equally important, it generates serendipity points through the diffusion and denoising processes conditioned on user portraits. Extensive experiments show that *SerenDiff* outperforms the state-of-the-art baseline models in predicting serendipity without sacrificing relevance. In addition, we analyzed the serendipity "zone" in user portraits to showcase the interpretability of *SerenDiff*.

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

## A  EXPECTEDNESS ESTIMATION

For the expectedness score, inspired by the computational method for calculating surprise levels proposed by the study Fu et al. (2023b), we define expectedness, the opposite of surprise, as the conditional likelihood of seeing an item given a user's history. Therefore, the level of the expectedness of item $i_m^u$ in $T^u$ at time $m$ is calculated as:

$$exp = p(i_m^u | T_{[1:m-1]}^u), \tag{13}$$

where the conditional likelihood $p(i_m^u | T_{[1:m-1]}^u)$ could be seen as a user's expectation of seeing the item $i_m^u$ give the history $T_{[1:m-1]}^u$. Using the Law of Total Probability and then the breakdown of a conditional likelihood, $p(i_t^u | T_{[1:m-1]}^u)$ can be re-written as:

$$p(i_m^u | T_{[1:m-1]}^u) = \sum_{i_h^u \in T_{[1:m-1]}^u} p(i_m^u | i_h^u) p(i_h^u | T_{[1:m-1]}^u)$$

$$= \sum_{i_h^u \in T_{[1:m-1]}^u} \frac{n(i_m^u, i_h^u)}{\sum_{i_m^u \in I_m} n(i_m^u, i_h^u)} p(i_h^u | T_{[1:m-1]}^u), \tag{14}$$

where $n(i_m^u, i_h^u)$ is the co-occurrence count for the item $i_m^u$ and an item $i_h^u$ in the user's history.

## B  IMPLEMENTATION DETAILS

The U-Net architecture used in *SerenDiff* consists of an encoder with three convolutional blocks (Conv2D with kernel size 3, stride 1, padding 1, followed by BatchNorm, GELU activation, and $2 \times 2$ MaxPooling) with channel dimensions [384, 768, 1536]; a bottleneck with two Conv2D layers ($1536 \rightarrow 3072$ and $3072 \rightarrow 1536$) each followed by BatchNorm and GELU; and a decoder consisting of four upsampling blocks with transposed convolutions, skip connections from the encoder, and Conv2D–GELU layers with channel dimensions [1536, 768, 384]. The network output is generated through a final Conv2D layer ($384 \rightarrow 384$) followed by a ReLU activation. We trained our models using the Adam optimizer.

To avoid the issue of non-smooth anisotropic distribution Huang et al. (2021) for the calculation of cosine similarity, we encoded item reviews to obtain item embeddings using Sentence-BERT (Reimers & Gurevych, 2019) and set the dimension of item embeddings 384. We set the learning rate 0.001, the dropout rate 0.2, and the regularizer decay 0.0001 for all the models.

To map continuous expectedness and relevance scores onto the 10×10 user portrait grid, we first quantize the scores into 10 discrete bins (10-quantiles) and then round each score to the corresponding bins. If multiple historical items fall into the same grid, we compute the average of their values and use this average as the "item" in that grid, ensuring that the information from all relevant items is preserved.

## C  TIME COMPLEXITY & TIME EFFICIENCY

Diffusion models typically contain millions of parameters and are computationally complex due to the iterative denoising process. Therefore the training and inference efficiency is another critical evaluation for *SerenDiff*. To reduce the burden of model training and inference, we precomputed and cached certain components of *SerenDiff*, such as item embedding calculation and user portrait construction.

We analyzed the time complexity of *SerenDiff* in comparison with the baseline models. The time complexity of select *SNPR* and *SerenEnhance* as representative serendipity-oriented models, *SAS-Rec* and *BERT4Rec* as representative relevance-oriented models, and *DiffRec* and *DreamRec* as representative diffusion-based recommendation models to baseline serendipity-oriented models (i.e.,

*SNPR* and *SerenEnhance*), relevance-oriented models (i.e., *SASRec* and *BERT4Rec*), and diffusion-based recommendation models (i.e., *DiffRec* and *DreamRec*) is $O(nd^2 + n^2d)$, where $n$ denotes the length of the user history $T^u$ and $d$ denotes the dimension size of the item embeddings. In contrast, our proposed *SerenDiff* has the time complexity of $O(L^2d^2)$, where $L$ is the width and height of the user portraits and $d$ is the dimension size of the item embeddings. In summary, *SerenDiff* has a comparable time complexity to other baselines.

To compare the time efficiency, we calculated the average training time of each model on 1 batch and the average inference time of each model on 1 instance. We trained and tested all the recommendation models on the *SerenLens-Book* dataset with 2 NVIDIA Tesla A100 GPUs. We kept all the hyperparameter settings the same for each model. The results are shown in Table 6. We observed that the training and inference times of *SerenDiff* are comparable to those serendipity or classic recommendation models, while being shorter than those diffusion-based recommendation models. These findings indicate the time efficiency of our *SerenDiff* in both training and inference.

Table 6: Model efficiency comparison

|  | $H_{seren}$@10 | training time per batch in seconds | inference time per instance in seconds |
|---|---|---|---|
| SNPR | 38.81% | 15.96s | 1.34s |
| SerenEnhance | 45.63% | 15.14s | 1.07s |
| SASRec | 41.37% | 10.20s | 0.78s |
| BERT4Rec | 41.79% | 12.56s | 0.88s |
| DiffRec | 40.23% | 17.33s | 3.19s |
| DreamRec | 41.02% | 17.69s | 3.98s |
| *SerenDiff* | 47.11% | 14.38s | 1.63s |

