# OpenReview forum: "SerenDiff: Generating Serendipity Recommendations through a Diffusion Model"
_ICLR.cc/2026/Conference — Submitted to ICLR 2026_

### Official Review · Reviewer_jDPy · 2025-10-31

**Soundness:** 3
**Presentation:** 2
**Contribution:** 2
**Rating:** 4
**Confidence:** 5

**Summary:**

The paper proposes SerenDiff, a novel method using a conditional diffusion model to generate serendipitous recommendations. Its key innovation is a 2D user portrait, constructed from axes of relevance and unexpectedness, which guides the generative process. This allows the model to create recommendations in a user's latent serendipity zone. Experiments on the SerenLens dataset show the model effectively identifies serendipitous items.

**Strengths:**

1. The core idea of framing serendipity as a conditional generative task using diffusion models is original. The 2D user portrait is a creative and effective conditioning mechanism.
2. The 2D portrait provides a clear visual and intuitive explanation for why a generated recommendation is considered serendipitous.
3. The model demonstrates state-of-the-art results on serendipity-specific metrics against a comprehensive set of baselines.

**Weaknesses:**

1. The evaluation relies on a single dataset . Findings may not generalize, as "serendipity" is highly context-dependent.
2. The paper is purely empirical. It lacks theoretical analysis on why the diffusion process's "creativity" effectively models the cognitive concept of "serendipity," or the properties of the proposed 10x10 portrait space.
3. The 10x10 portrait is small. The complexity is a concern if finer granularity (a larger portrait) is needed, but this trade-off isn't explored.
4. The portrait's quality is highly dependent on the models used to calculate expectedness and item embeddings. The impact of these upstream choices is not analyzed.

**Questions:**

1. What is the model's performance and efficiency sensitivity to the portrait size?
2. Which specific model was used to calculate the crucial expectedness score used to build the portrait?
3. To ensure reproducibility, would the authors be willing to provide the code？

---

> ### Author Response · Authors · 2025-11-28
>
> Response to Question 1: Thank you for the insightful suggestion. To further examine stability and sensitivity, we conducted additional experiments by enlarging the window size of user portraits to 100×100, while keeping the 384-dimensional channels. As shown in Table 3 of the revised manuscript, it can be observed that training remained stable and achieved a slightly higher performance than the 10×10 setting. However, this came at a substantial computational cost. Compared with the 10×10 setting, the performance gain was marginal while the efficiency dropped sharply. These results indicate that although the diffusion framework can handle the high channel dimensionality, scaling up the portrait size yields diminishing returns relative to the increased computational overhead.
>
> Response to Question 2: We thank the reviewer for the insightful comment. As discussed in Section 4.2, the expectedness scores of user portraits are calculated by co-occurrence-based expectedness estimation, which is inspired by [1]. The detailed calculation process of the expectedness scores can be found in Section 4.2. We have also added more calculation details in Appendix A in the revised manuscript.
>
> Response to Question 3: We appreciate the reviewer’s comment. As mentioned in the paper, we plan to publicly release the full source code and the corresponding datasets on GitHub after the paper is accepted.
>
> Response to Weakness 1: We thank the reviewer for this important point. We acknowledge that serendipity is context-dependent, and evaluating on a single dataset may limit generalizability.  However, as discussed in Section 5.1, there are currently only two publicly available serendipity ground-truth datasets with user history. Among the two datasets, the SerenLens dataset provides a relatively large and reliable scale of ground-truth data in both the book and movie domains. Accordingly, we adopt the SerenLens dataset in our paper. In addition, although our evaluation is conducted on a single dataset, we believe our findings provide meaningful insights into modeling and generating user-aware serendipity, and the methodological contributions are broadly applicable to other domains. In future work, we plan to validate SerenDiff on additional datasets to further assess generalizability.
>
> Response to Weakness 2:  We thank the reviewer for this valuable comment. We agree that a theoretical analysis would further strengthen the work. Our two-dimensional user portrait enhances traditional history modeling by explicitly capturing unexpectedness and relevance, thereby enabling more nuanced preference modeling. The diffusion-based approach and the two-dimensional user portrait are complementary. The diffusion model provides strong inference capability and generative “creativity,” allowing it to propagate user preferences across related items. The portrait ensures that this propagation occurs along the dimensions of unexpectedness and relevance, guiding the generation process toward psychologically meaningful serendipity. This synergy allows SerenDiff to balance unexpectedness and relevance for high-quality serendipitous recommendations. We view this as a first step toward a more formal theoretical analysis and plan to explore deeper connections between cognitive serendipity and diffusion dynamics in future work.
>
> Response to Weakness 3: Please refer to the response to Question 1.
>
> Response to Weakness 4: We thank the reviewer for the insightful comment. We agree that choosing different upstream models for expectedness estimation and item embeddings may influence performance. To align with our definition of unexpectedness and rely on users’ historical data, we adopt the well-established co-occurrence-based expectedness estimation method [1]. For relevance estimation, we use the widely-adopted Sentence-BERT model [2] to avoid issues associated with the non-smooth, anisotropic distribution of embeddings observed in cosine-similarity calculation [3]. We are open to exploring alternative upstream model choices. However, our focus is the "downstream" model: a diffusion model to generate serendipity.
>
>
> References
> [1] Fu, Z., Niu, X. and Yu, L., 2023, July. Wisdom of crowds and fine-grained learning for serendipity recommendations. In Proceedings of the 46th International ACM SIGIR Conference on Research and Development in Information Retrieval (pp. 739-748).
>
> [2] Reimers, N. and Gurevych, I., 2019, November. Sentence-BERT: Sentence Embeddings using Siamese BERT-Networks. In Proceedings of the 2019 Conference on Empirical Methods in Natural Language Processing and the 9th International Joint Conference on Natural Language Processing (EMNLP-IJCNLP) (pp. 3982-3992).
>
> [3] Huang, J., Tang, D., Zhong, W., Lu, S., Shou, L., Gong, M., Jiang, D. and Duan, N., 2021, November. WhiteningBERT: An Easy Unsupervised Sentence Embedding Approach. In Findings of the Association for Computational Linguistics: EMNLP 2021 (pp. 238-244).

---

### Official Review · Reviewer_WGA2 · 2025-11-01

**Soundness:** 3
**Presentation:** 2
**Contribution:** 2
**Rating:** 4
**Confidence:** 4

**Summary:**

This work propose SerenDiff, a diffusion-based recommendation model to address the challenge of generating serendipitous recommendations. It claims to be the first work to formulate serendipity recommendation as an item generation task, where a two-dimensional user portrait based on "expectedness" and "relevance" is constructed and a conditional diffusion model is used to denoise the recommended item generation. Emprically, the proposed method achieved competitive performance on balancing serendipity and relevance metrics.

**Strengths:**

- Compared to previous work on diffusion model for recommendation, the work provides an interesting perspective to construct user profile by explicitly decomposing serendipity into expectedness and relevance.
- The two-dimensional user portrait enhances model interpretability as shown in the case studies, and the ablation study confirms its utility.

**Weaknesses:**

- Missing technical details to fully understand the proposed method and reproducibility (see more in Questions)
- Technical novelty lies more in the creation of user portrait with interaction data, but less in understanding why the diffusion process is suited (or needs to be adapted) for discovering serendipity beyond its general generative capabilities.
- Some nuanced issues in evaluation setting, which might weaken the validity of the results (see more in Questions)

**Questions:**

- In the User Portrait Construction, the portrait is a 10x10 grid. How are continuous expectedness and relevance scores quantized onto this discrete grid? What happens if multiple historical items fall into the same grid?
- The calculation of unexpectedness is based on the conditional likelihood of seeing an item given a user's history, but such calculation tends to suffer from exposure bias. Is any debiasing techniques used is ensure robustness to data bias, etc?
- The channel dimension of the "user portrait image" is quite high compared to standard images, as it correspond to the item embedding dimension (in this case, 384 as indicated in Appendix sec. A). How is diffusion training affected by the selection of this dimension? Can you discuss more about the stability of training when scaling up this dimension?
- In evaluation, several choice needs to be more careful. For example, sampled evaluation is used in this paper, but has shown to be misleading by [1]; cross-validation is used to create train/test splits, while in recommendation this type of random splits evaluation might cause leakage [2], where temporal based splits might be preferred. More rigorous evaluation is needed to better justify the empirical results.


[1] Walid Krichene and Steffen Rendle. 2022. On sampled metrics for item recommendation.

[2] Zaiqiao Meng, Richard McCreadie, Craig Macdonald, and Iadh Ounis. 2020. Exploring Data Splitting Strategies for the Evaluation of Recommendation Models.

---

> ### Author Response · Authors · 2025-11-28
>
> Response to Question 1: Thank you for the question. To map continuous expectedness and relevance scores onto the 10×10 user portrait grid, we first quantize the scores into 10 discrete bins (10-quantiles). If multiple historical items fall into the same grid, we compute the average of their values and use this average as the "item" in that grid, ensuring that the information from all relevant items is preserved.
>
> Response to Question 2: Thank you for raising this important point. We acknowledge that estimating unexpectedness based on conditional likelihoods can be affected by exposure bias. To mitigate this issue, we filtered out users with fewer than 10 historical items, retaining only those with sufficient history to obtain a more reliable expectedness estimate. Also, part of the reason that we used diffusion models is that they do not rely on a "complete" user history. The models can generate the missing part based on partial user history. Nonetheless, we agree that there might still be exposure bias. A debiasing strategy is an open research question, not only for this paper, but also for many recommendation studies that rely on a user history.
>
> Response to Question 3: We thank the reviewer for the comment. Indeed, the channel dimension of the user portrait (384, corresponding to the item embedding size) is higher than typical image channels. In practice, we found that diffusion training remains stable at this dimensionality because the portrait is sparse and structured, allowing the model to efficiently capture dependencies without excessive noise accumulation. To further examine stability and sensitivity, we conducted additional experiments by enlarging the window size of user portraits to 100×100, while keeping the 384-dimensional channels. As shown in Table 3 in the revised manuscript, it can be observed that training remained stable and achieved a slightly higher performance (49.59% on $H_{seren}@10$) than the 10×10 setting (47.11% on $H_{seren}@10$). However, this came at a substantial computational cost: training time increased to 248 seconds per batch, and inference time to 4.13 seconds per instance. Compared with the 10×10 setting, the performance gain was marginal while the efficiency dropped sharply. These results indicate that although the diffusion framework can handle the high channel dimensionality, scaling up the portrait size yields diminishing returns relative to the increased computational overhead.
>
> Response to Question 4: We appreciate the reviewer’s comments on evaluation. Sampled evaluation is widely used in large-scale recommendation settings where full-ranking is infeasible. We applied the same set of samples across all models, ensuring fair comparison. Since our serendipity recommendation task is not a sequential recommendation task, a temporal-based split is not relevant. Instead, a user-based split (some users in training and other users in test) is more appropriate.

---

### Official Review · Reviewer_nF6o · 2025-11-01

**Soundness:** 2
**Presentation:** 2
**Contribution:** 2
**Rating:** 2
**Confidence:** 4

**Summary:**

This paper proposes a serendipity diffusion recommendation model (SerenDiff), to generate serendipity recommendations leveraging diffusion model. It regards the user historical behaviors as an image, and the serendipity recommendation generation as a recovering process of the corrupted image. Extensive experiments and analyses have demonstrated its effectiveness and interpretation.

**Strengths:**

1. The investigated question is interesting and valuable.
2. Results in Table 1 proves that SerenDiff generally outperforms existing baselines under the Serendipity-Oriented Experiments, achieving statistically significant gains in most columns.
3. The case study is easy-to-understand, it visualizes the distances and directions of these serendipitous movies compared to users’ typical reach, offering a form of interpretation.

**Weaknesses:**

1. Could the authors clarify the technical contributions of the paper? As conditional diffusion models have already been widely applied in recommender systems, the contribution from this aspect appears to be marginal.
2. The description of the method is not very clear. For instance, what are the dimensions of M^u and M^u_{\text{seren}} mentioned in Sections 4.1 and 4.3, and what do they represent? The authors bolded these symbols in the figures but not in the main text, which may cause confusion for future readers.
3. I’m concern about the authors’ claim that their method significantly outperforms 6 out of 9 baselines according to a two-tailed t-test.
4. Among the baselines used for comparison, only SerenPrompt and DreamRec are from 2024, while all the others are before 2023. Besides, the compared sequential recommenders (BERT4Rec and SASRec) are relatively early SR works. However, even so, they still significantly outperform SerenDiff in the relevance-oriented experiments.
5. Lack of the statistics of the utilized datasets.
6. Lack of the source code and corresponding datasets, and the author promise to release the source code on GitHub after the paper is accepted.

**Questions:**

Please refer to the weakness

---

> ### Author Response · Authors · 2025-11-28
>
> Response to Weakness 1: We thank the reviewer for the comment. While conditional diffusion models have been applied in recommender systems, SerenDiff makes distinct technical contributions by explicitly modeling serendipity (rather than relevance): 1) It models users’ openness and acceptance zones to better capture individual preferences. 2) It is the first effort that treats serendipity as user-aware creativity, increasing the likelihood of psychological acceptance by recommending items slightly beyond users’ typical choices. 3) It provides visualized explanations, enhancing transparency and building user trust. As discussed in Section 1, modeling serendipity in recommender systems is challenging due to its elusive nature. Another key challenge is interpretability, as many existing serendipity recommendation models lack transparency in revealing their inner workings. SerenDiff leverages diffusion models to address both challenges. Diffusion models, as generative models based on corrupting and denoising processes, are particularly suitable for modeling creativity, as they can generate new and unique data beyond the original training set.
>
> Response to Weakness 2: Thank you for your comments. As described in Section 4.1, $M^u$ refers to the two-dimensional user portrait capturing the user’s interaction history, while $M^u_{seren}$ denotes the generated serendipity representation derived from user history. Section 5.4 further clarifies that both matrices share the same dimensionality, with a size of 10×10×number of channels. Also, we have unbolded the symbols $M^u$ and $M^u_{seren}$ in Figure 3 and Figure 4 of the revised manuscript to match the text.
>
> Response to Weakness 3: We appreciate the reviewer’s comment. We argue that requiring statistically significant improvements over all state-of-the-art baselines is too strict, as certain baselines have inherent advantages in specific evaluation metrics. For instance, as discussed in Section 6.2, SerenEnhance employs a pairwise learning objective with a strong discriminative capability, enabling it to achieve superior top-1 book recommendation performance compared with our method. Therefore, we believe that demonstrating statistical significance over the majority of competitive baselines is good enough to show our model’s is generally better. In fact, it is common practice to to do so (e.g., [1]).
>
> Response to Weakness 4: We thank the reviewer for this insightful comment. While it is true that most of our baselines are from before 2023, and that BERT4Rec and SASRec are earlier sequential recommendation models, we included them because they are widely adopted and recognized as strong benchmarks for both relevance- and serendipity-oriented evaluations. Their inclusion ensures that our comparisons remain meaningful and interpretable. We would also like to emphasize that SerenDiff is explicitly designed to optimize serendipity, rather than only relevance. Therefore, some reduction in relevance-oriented metrics is expected and reflects a tradeoff: our model aims to balance serendipity and user acceptance, which is the primary focus of our work, as also demonstrated in [1], [2], and [3].
>
> Response to Weakness 5: Thank you for the suggestion. As introduced in Section 5.1, we use the SerenLens dataset [1], which was collected from crowd workers’ annotations on review writers’ serendipity experiences. It contains 265,037 annotated book reviews and 74,967 annotated movie reviews. We added a table (Table 1) in the revised manuscript for the key statistics of the SerenLens dataset.
>
> Response to Weakness 6: We appreciate the reviewer’s concern. As mentioned in the paper, we plan to publicly release the full source code and the corresponding datasets on GitHub after the paper is accepted. This will ensure reproducibility and allow the community to build upon our work. We will also provide clear instructions and documentation to facilitate usage. The details and the link of the SerenLens dataset can be found in [1]'s paper.
>
> References
>
> [1] Fu, Z., Niu, X. and Yu, L., 2023, July. Wisdom of crowds and fine-grained learning for serendipity recommendations. In Proceedings of the 46th International ACM SIGIR Conference on Research and Development in Information Retrieval (pp. 739-748).
>
> [2] Li, X., Jiang, W., Chen, W., Wu, J., Wang, G. and Li, K., 2020, April. Directional and explainable serendipity recommendation. In Proceedings of The Web Conference 2020 (pp. 122-132).
>
> [3] Zhang, M., Yang, Y., Abbas, R., Deng, K., Li, J. and Zhang, B., 2021, October. SNPR: A serendipity-oriented next POI recommendation model. In Proceedings of the 30th ACM International Conference on Information & Knowledge Management (pp. 2568-2577).

---

### Official Review · Reviewer_u6GH · 2025-11-10

**Soundness:** 3
**Presentation:** 3
**Contribution:** 3
**Rating:** 6
**Confidence:** 3

**Summary:**

The text introduces SerenDiff, a novel serendipity diffusion recommendation model that addresses the challenge of modeling and interpreting serendipity by adapting a conditional diffusion model for item generation. The core novelty involves transforming a user’s one-dimensional history sequence into a two-dimensional, image-like user portrait that explicitly represents the trade-off between the two essential facets of serendipity: unexpectedness and relevance. SerenDiff then leverages diffusion models, treating recommendation as a process of recovering a corrupted “image”, to generate serendipity points corresponding to hidden user demands.

**Strengths:**

1. The methodology is well motivated. The generative nature of diffusion models is well-suited for exploring the latent space beyond immediate user relevance.
2. Transforming 1D user interaction sequences into 2D "images" based on expectedness and relevance is a novel approach. It effectively translates a standard recommendation problem into an "image" generation problem.
3. The 2D user portrait provides good interpretability.

**Weaknesses:**

1. A large number of time steps ($T=1,000$) in SerenDiff is inefficient compared to classic recommendation models, as shown in Appendix.
2. The number of time steps is a key parameter in diffusion models, and a hyperparameter study would help determine its optimal value.

**Questions:**

In image generation tasks, diffusion models typically use around 1000 steps. However, in recommendation tasks, models such as DiffRec only use 5–10 steps. Is it necessary to use as many as 1000 steps in this setting? Moreover, how does SerenDiff achieve inference speed only about twice as slow as SASRec despite using 1000 steps? Any tradeoff between efficiency and performance when choosing the number of time steps?

---

> ### Author Response · Authors · 2025-11-28
>
> Response to Question 1: Thank you for the question. Existing diffusion-based recommender models (e.g., [1], [2]) use only 5-10 denoising steps because they generate low-dimensional latent vectors or very small matrices. In contrast, our model generates a substantially higher-dimensional image-like matrix that encodes the serendipity structure in a user’s history. This higher structural complexity requires more gradual denoising to achieve stable and consistent generation. To more thoroughly examine how diffusion steps influence our framework, we added additional studies to assess their impact on both performance (measured via $H_{seren}@10$) and efficiency (training/inference latency) in the revised manuscript. The detailed results can be found in Table 2 of the newly revised manuscript. Overall, we find that increasing the number of diffusion steps steadily improves serendipity performance, while the associated increase in computational cost remains moderate and manageable.
>
> Response to Question 2: Thank you for the question. Although our image-like user portraits are high-dimensional, they are highly sparse and structured, which allows the denoising operations to be computed efficiently. As a result, the per-step computational cost is low, enabling SerenDiff to achieve inference speed only about twice as slow as SASRec even with 1000 steps.
> Regarding the trade-offs between efficiency and performance, there is indeed a balance: reducing the number of steps improves inference speed but may slightly degrade the quality of the generated serendipity patterns. To clarify this, we included an ablation study in the revised paper that reports both performance and inference time under different numbers of diffusion steps, illustrating the efficiency-performance trade-off.
>
> References
>
> [1] Wang, W., Xu, Y., Feng, F., Lin, X., He, X. and Chua, T.S., 2023, July. Diffusion recommender model. In Proceedings of the 46th international ACM SIGIR conference on research and development in information retrieval (pp. 832-841).
>
> [2] Li, Z., Sun, A. and Li, C., 2023. Diffurec: A diffusion model for sequential recommendation. ACM Transactions on Information Systems, 42(3), pp.1-28.

---

### Meta-Review · Area_Chair_93JN · 2026-01-09

**Summary:**

This manuscript introduced a diffusion-based serendipity recommendation and conducted some experiments to validate the proposed model. This manuscript has clear limitations regarding limited technical novelty, insufficient experiment validation.

**Reviewer Concerns:**

Most concerns were not addressed.

**Reviewer Scores:**

Unlike to change.

---

### Decision · Program_Chairs · 2026-01-26

Reject